# Unified recurrent neural network for many feature types

## Abstract

There are time series that are amenable to recurrent neural network (RNN) solutions when treated as sequences, but some series, e.g. asynchronous time series, provide a richer variation of feature types than current RNN cells take into account. In order to address such situations, we introduce a unified RNN that handles five different feature types, each in a different manner. Our RNN framework separates sequential features into two groups dependent on their frequency, which we call sparse and dense features, and which affect cell updates differently. Further, we also incorporate time features at the sequential level that relate to the time between specified events in the sequence and are used to modify the cell's memory state. We also include two types of static (whole sequence level) features, one related to time and one not, which are combined with the encoder output. The experiments show that the proposed modeling framework does increase performance compared to standard cells.

## 1 Introduction

The study of time series has a long history and the literature for it covers many different methods (Hamilton (1994)). The study of asynchronous time series is an important subset of this. Asynchronous time series are series for which features are sampled at irregular time intervals, and at any given time step new values of any subset of features may be present. When a feature does not change values often it can be treated as being present only at times of change. For example, in industrial IoT several sensors can be monitored, each one with its own sampling rate. We can have a feature that corresponds to the production rate that seldom changes and it impacts predictions. This clearly presents difficulties from both a modeling and data input perspective. Fixed sampling, repeating missing values, and other types of data imputation have all been used to varying degrees of success. More recent attempts using machine learning for asynchronous time series use Gaussian processes (Cunningham et al. (2012); Li & Marlin (2016)), which are hard to scale in the presence of many features.

Deep learning is a much more recent development, and it is achieving great success in many fields, e.g. machine vision (Krizhevsky et al. (2012); Szegedy et al. (2015); He et al. (2016)) and natural language processing (Wu et al. (2016); Yin et al. (2017)). Recurrent neural networks (RNNs) are a type of a neural network that use shared weights applied to sequences. These types of networks achieve state of the art performance for sequential data. Time series can be thought of as sequences in the RNN context, and thus they are amenable to RNN-based solutions (Malhotra et al. (2015); Che et al. (2018); Längkvist et al. (2014)). However, many of the most widely used RNN cells, e.g. the Long Short Term Memory (LSTM) cell (Hochreiter & Schmidhuber (1997)), are built for problems where time either does not matter or there is a constant time step size.

The adaptation of these cells for asynchronous sequences is a more recent development, varying from combining traditional methods with deep learning (Binkowski et al. (2018); Wen et al. (2017)) to pure neural network approaches (Neil et al. (2016); Baytas et al. (2017)). These attempts may take into account time between samples and the patterns of missing values, but they do not fundamentally treat features that are present with varying levels of frequency any differently. Thus, a feature that occurs at every time step and a feature that almost never occurs at any given time step update the cell in the same manner.

We address this shortcoming by introducing a new recurrent cell that uses features that are present frequently, called dense features, differently from features that are rarely present, termed sparse features, when updating the cell's hidden and memory states. In particular, our cell's hidden state is split into two parts, one part for **dense** features and one part for **sparse** features. Additionally, each sparse feature maintains its own hidden and memory state that are updated only at time steps when that feature is present. Thus, the update for the sparse part of the overall hidden state depends on the subset of sparse features present at any given time step, while the update for the dense part of the overall hidden state happens in the standard LSTM manner.

In addition to handling these two feature types, our cell also accounts for the irregular time between time steps, expanding on the work of Baytas et al. (2017) by allowing more flexibility for ingesting time features. Specifically, we modify the decay function to handle an arbitrary number of **decay** features, not just the time elapsed since the last time step. The model also permits sequence level features, distinguishing time related features, called **static decay** features, which are handled in the same manner as within the recurrent cell, from the non-time related features, called **static standard** features, which are embedded and concatenated with the sequence output in the standard manner. Overall, we support five feature types.

The main contributions of this work are as follows. First, the introduction of a new recurrent cell that accounts for asynchronous feature sampling and treats features in a variable manner dependent on the frequency with which their values are present. Second, an expansion of the time-aware LSTM (TLSTM) to handle multiple time features, and the extension of this method applied to the output of the recurrent cell itself. Third and finally, a general recurrent framework to incorporate the five aforementioned feature types.

## 2 RELATED WORK

When dealing with an irregularly sampled multivariate sequence, one can consider the variables not present at any given time step to be missing. Prior work done on recurrent networks with missing data includes (Bengio & Gingras (1996); Tresp & Briegel (1998)). Lipton et al. (2016) test the performance of different types of imputation and missing value indicator input features. This work inputs the missing values patterns into the recurrent network by concatenating with the feature vectors while using a standard LSTM network. They find zero filling missing features and using indicator features outperforms data imputation, which may actually interfere with the network's ability to pick out missing values.

Che et al. (2018) investigate a similar approach and improve upon this by modifying a GRU (Cho et al. (2014)) cell to separately incorporate the missing value patterns as well as the time between successive measurements for each feature. It does this by introducing two decay mechanisms. The first decay acts to bring a missing value toward the baseline mean value from its last measured value so the old value is not input repeatedly. This work treats all features in the same manner, regardless of the frequency a missing value is present for any given feature. The work in Che et al. (2018) does not differentiate between features that never have missing values and features where almost all values are missing. The second decay mechanism uses the time feature to decay the hidden state before calculating the new hidden state. We note that missing data at a time is not the same as features not being present (sampled) or a feature rarely changing value. We focus on the latter while Che et al. (2018); Bengio & Gingras (1996); Tresp & Briegel (1998) focus on the former.

Baytas et al. (2017) incorporate the time between events for event based sequences with irregular intervals in a similar fashion to the second decay mechanism in Che et al. (2018). This is done by decomposing the memory state into short and long term components, and then multiplying the short term component by a decay factor that decreases as the time between events increases. We use this in our own recurrent cell, and also expand upon the idea by incorporating multiple time features, not just limited to the time between adjacent events. We also use the same decomposition and decay method outside of our recurrent cell since the prediction time occurs some time after the final event. Their work does not consider sparse features.

Neil et al. (2016) modify the standard LSTM cell to address its issues with asynchronous time series. They do this by incorporating a time gate into the network that controls when and to what extent the hidden and memory states can be updated. This means that there are time steps where no updates are

made, and so the network accumulates changes in the feature vector before updating the hidden and memory states. While the time gate has learnable parameters, they still do not differentiate between features that occur at every time step versus those that rarely occur in the cell updates.

In summary, there is no prior work that considers the notion of sparse features and a single model for all five feature types.

## 3 MODELS

The main contribution of this paper is the development of a recurrent cell which handles five distinct feature types. The following subsections review each of the feature types and describes how they are handled in the Sparse Time LSTM (STLSTM model). The five types are referred to as the following: dense features, sparse features, decay features, static standard features, and static decay features. The dense features are the standard RNN sequence input, such as the embedding of an event type or the sequence state features at a given time step, and are present at any given time in a sequence. The sparse features change values infrequently, and thus they can be considered as being present only at select times. Stating that the value is not present means the feature value has not changed from the previous time. A further assumption is that these feature values do not decay with time. The decay features must be related to time measured with respect to some condition, such as time since the previous event in an event driven sequence. These features are particularly useful when events in the sequence are clustered, meaning that we observe many events in close proximity alternating with long gaps. Both the static standard and the static decay features give information about the sequence as a whole, rather than being tied to a specific time step. Static standard features are time invariant, e.g. the gender of a sequence's object, whereas static decay features are time related, e.g. the time since last event in the sequence was observed. Figure 1 depicts these concepts.

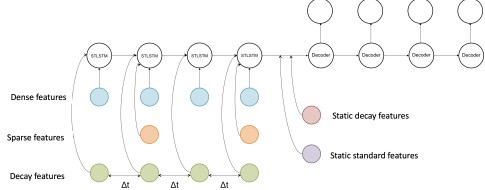

Figure 1: The framework of the five feature types.

Figure 2: The memory state is decomposed into short and long term components, and then the standard LSTM updates are performed.

We consider a neural network to be a mapping $f(X)$, where $f$ represents the network and $X$ is the input to the network. We split $X$ first into two parts, $X = (x^{static}, x)$ where $x^{static}$ represents the sequence level static features and $x$ represents the sequence features. The static features can be split into two parts $x^{static} = (x^{static,s}, x^{static,\Delta})$, for static standard and static decay features respectively. The sequence features can further be broken down into three parts, $x = (x^d, x^\Delta, x^{sp})$ that represent the dense, decay, and sparse features respectively. For notational convenience, each of these features has a vector at each time step of the sequence (even though in our implementation the sparse features would be used only when the value changes). The dense features are denoted as $x^d = (x_1^d, x_2^d, ..., x_T^d)$ for a sequence with $T$ time steps. The decay features are represented as $x^\Delta = (x_1^\Delta, x_2^\Delta, ..., x_T^\Delta)$. The sparse features, however, are represented slightly differently than the previous two types of features, because their changes are not always present. We use the term *present* for a sparse feature when its value changes. A single sparse feature $k$ can be represented as a sequence of tuples, $x_k^{sp} = ((m_{1k}, x_{1k}^{sv}), ..., (m_{Tk}, x_{Tk}^{sv}))$, where $m_{tk} \in \{0, 1\}$ is a mask value that is 1 when the feature is present and 0 if the feature is missing. If $m_{tk} = 1$, then $x_{tk}^{sv}$ is the actual new feature value, and otherwise the value of $x_{tk}^{sv}$ does not impact the feed forward computation. If a sparse feature is present at a time, then this input is passed into and used in the cell updates. If there is no sparse input at a given time step, the sparse part of the cell is untouched at that time step.

Our recurrent cell is built off of the commonly used LSTM Hochreiter & Schmidhuber (1997), as well as its time-aware variant (TLSTM) Baytas et al. (2017). For LSTM-type cells, there is a memory state, $C$, and a hidden state, $h$, which pass from the cell at the previous time step and into the cell along with the features at the current time step. The memory state boosts performance for

long term dependencies, and the gate structure of the LSTM allows selective parts of the memory to be forgotten and updated. The standard LSTM architecture does not address irregularly sampled events, however, and the TLSTM variant addresses this by using the time elapsed between events to modify the memory state.

In the TLSTM cell, at each time step, first a fully connected layer is used to decompose the cell's memory state into short and long term components. The time elapsed is passed as the input to a non-increasing decay function that maps to a scalar decay factor. The short term component of the memory state is multiplied by the decay factor that decreases as more time passes. The equations for a single decay feature $x_t^\Delta$ for this process are below on the left, where $g$ is a decay function, $x_t^\Delta$ is the decay input, and $C_{t-1}$ is the previous cell memory state.

After the memory state is modified, the cell update is performed in the standard LSTM manner, with the exception that the modified cell memory state calculated in Equation 1 is used to update the current memory state. The equations are as follows on the right where $x_t^d$ is the feature vector to the current time step. A diagram of this base cell is given in Figure 2.

$$C_{t-1}^S = \tanh\left(W^\Delta C_{t-1} + b^\Delta\right)$$

$$f_t^d = \sigma(W_{fh}^d h_{t-1} + W_{fx}^d x_t^d + b_f^d)$$

$$\hat{C}_{t-1}^S = C_{t-1}^S \cdot g\left(x_t^\Delta\right)$$

$$i_t^d = \sigma(W_{ih}^d h_{t-1} + W_{ix}^d x_t^d + b_i^d)$$

$$C_{t-1}^L = C_{t-1} - C_{t-1}^S$$

$$\tilde{C}_t^d = \sigma(W_{Ch}^d h_{t-1} + W_{Cx}^d x_t^d + b_C^d)$$

$$C_{t-1}^* = C_{t-1}^L + \hat{C}_{t-1}^S,$$

$$C_t^d = f_t^d * C_{t-1}^* + i_t^d * \tilde{C}_t^d,$$

$$o_t^d = \sigma(W_{oh}^d h_{t-1} + W_{ox}^d x_t^d + b_o^d)$$

$$h_t^d = o_t^d * tanh(C_t^d)$$

### 3.1 DENSE FEATURES

The dense features are the standard features used for recurrent models. It is assumed that the features are not time decay related, e.g. they encode attributes of events, but not times between events. At each time step, the hidden state from the previous time step, the memory vector, and the feature vector for the current time step are input into the cell, and the hidden state and memory vector are updated for the next time step.

### 3.2 DECAY FEATURES

Decay features are not relevant or applicable for some RNN applications such as text or video, but for a sequence with measurements or events that occur with varying frequency, time information can add predictive power.

In the beginning of this section, a model expressed by Equation 1 is presented that allows a single decay feature. Here we extend this to multiple decay features. Incorporating multiple decays requires a simple modification to the decay function $g$. Instead of mapping from a scalar to a scalar, it maps from a vector to a scalar. The proposed $g$ function is $g(x_t^\Delta) = \frac{1}{\log\left(e + \alpha^T x_t^\Delta\right)}$, where $\alpha \geq 0$ is a trainable vector.

### 3.3 SPARSE FEATURES

Sparse features are similar in nature to the dense features, with the exception that their values change very rarely throughout the sequence. Like the dense features, the sparse features describe the state of the sequence at a particular time step. However, since they rarely change from one step to another, when treated in the same manner as dense features, the sparse features would just repeat inputs the vast majority of time steps. There is no clear cutoff point in new value frequency for which a dense feature turns into a sparse feature.

We propose a new recurrent cell, called the Sparse Time Long Short Term Memory (STLSTM) cell, that attempts to more strongly capture the change in sparse features. The STLSTM cell has a hidden state split into two parts, one corresponding to the dense features, and the other corresponding to the sparse features, $h_t = (h_t^d, h_t^{sp})$. Further, each sparse feature has its own memory state, and this memory state is only updated when there is a change in the feature value. At each time step, the

dense part of the hidden state is updated based on Equation 1 and Equation 2. For the sparse part of the hidden state, there is a proposed hidden state from each sparse feature, and these proposed hidden states are aggregated together.

Let us denote the memory state of sparse feature $k$ as $C_{tk}^{sp}$ and the corresponding hidden state as $h_{tk}^{sp}$. The main idea is to not change $C_{tk}^{sp}$ if $m_{tk} = 0$, i.e. to simply carry it over, and otherwise if $m_{tk} = 1$ to use the feature $x_{tk}^{sv}$ within LSTM-like update equations. Formally, if $m_{tk} = 0$, then there is no change to the memory state or the hidden state, that is,

$$C_{tk}^{sp} = C_{t-1,k}^{sp}, \quad h_{tk}^{sp} = h_{t-1,k}^{sp}.$$

If there is a change in the feature value ($m_{tk} = 1$), the new memory state is determined by the following equations:

$$
\begin{aligned}
f_{tk}^{sp} &= \sigma(W_{fh}^{sp} h_{t-1} + W_{fx}^{sp} x_{tk}^{sv} + b_f^{sp}) \\
i_{tk}^{sp} &= \sigma(W_{ih}^{sp} h_{t-1} + W_{ix}^{sp} x_{tk}^{sv} + b_i^{sp}) \\
\tilde{C}_{tk}^{sp} &= \sigma(W_{Ch}^{sp} h_{t-1} + W_{Cx}^{sp} x_{tk}^{sv} + b_C^{sp}) \\
C_{tk}^{sp} &= f_{tk}^{sp} * C_{t-1,k}^{sp} + i_{tk}^{sp} * \tilde{C}_{tk}^{sp}.
\end{aligned}
$$

Once the memory state is calculated, the hidden state parts are calculated using standard output gates. The equations for this are as follows:

$$
\begin{aligned}
o_{tk}^{sp} &= \sigma(W_{oh}^{sp} h_{t-1} + W_{ox}^{sp} x_{tk}^{sv} + b_o^{sp}) \\
h_{tk}^{sp} &= o_{tk}^{sp} * tanh(C_{tk}^{sp}) \\
h_t &= \left[ h_t^d, \mathcal{L}(h_{t1}^{sp}, ..., h_{tm}^{sp}; W_{ah}) \right] \\
o_t &= \left[ o_t^d, \mathcal{L}(o_{t1}^{sp}, ..., o_{tm}^{sp}; W_{ao}) \right]
\end{aligned}
$$

Here $\mathcal{L}$ is an aggregation function that yields a vector of the same dimension as $h_{tk}^{sp}$ and $W_{ah}, W_{ao}$ are trainable parameters. For example, $\mathcal{L}$ can be simply averaging the hidden state proposals or we can place a fully connected layer over all of the sparse hidden states, and use this layer to compute a single hidden state for all sparse features. A diagram for the STLSTM cell is depicted in Figure 3.

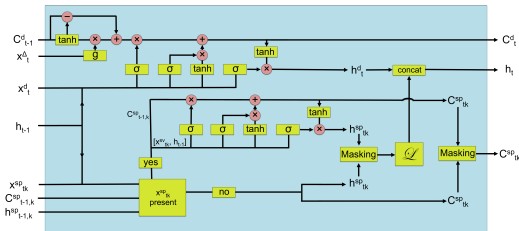 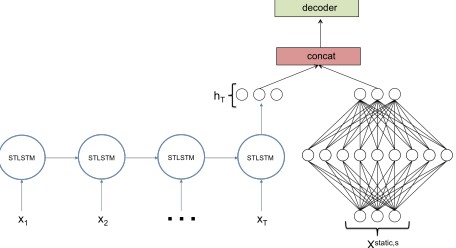

Figure 3: Full wiring for the STLSTM cell. The sparse feature update follows one of two possible paths dependent on the masking.

Figure 4: Static features are embedded and then concatenated to RNN output.

### 3.4 STATIC STANDARD FEATURES

All previous discussions dealt with features that can be assigned to a specific time step in a sequence, but we can also have features that are sequence level instead of time step level. These features, which are not time related, give information about the sequence as a whole, so they should not serve as inputs to the recursive part.

The incorporation of these features is fairly simple. First the features are passed through a fully connected layer to get an embedding, and then this embedding is concatenated with the output, $h_T$, of the recurrent cell at the final time steps. A diagram of this can be seen in Figure 4.

### 3.5 STATIC DECAY FEATURES

Similar to the static standard features, the static decay features apply to the sequence as a whole rather than an individual time step. Their values are related to the time since the last observed event, e.g. the time between the final time step and the prediction time.

These features are used in the same manner as the decay features within the STLSTM cell. The output of the recurrent network, $h_T$, is decomposed into short and long term components, and the decay factor is applied to the short term component before being added back to the long term component. Formally, this is given as

$$
\begin{aligned}
h_T^S &= \tanh\left(W^{static,\Delta}h_T + b^{static,\Delta}\right) \\
\hat{h}_T^S &= h_T^S \cdot g\left(x^{static,\Delta}\right) \\
h_T^L &= h_T - h_T^S \\
h_T^* &= h_T^L + \hat{h}_T^S
\end{aligned}
$$

This modified output is then concatenated with the embedding of the static standard features before input to the decoder.

## 4 COMPUTATIONAL STUDY

For all experiments in this section we use Keras with a Tensorflow backend on a single GeForce GTX 1080 GPU card. We use ADAM as the optimization algorithm, and keep track of the validation F1 score to control the number of training epochs. If the validation F1 score does not increase for 15 epochs, then training is terminated. We use standard weight initialization, Glorot uniform for weights connected to the inputs and orthogonal initialization for the recurrent weights. To test out the STLSTM cell with all five feature types, we use two data sets, one public and one proprietary.

### 4.1 POWER CONSUMPTION DATA SET

The public data set is a modified version of the UCI household electric power consumption dataset (Dheeru & Karra Taniskidou (2017)). This data set contains measurements for seven different electrical quantities and sub-metering values with a sampling rate of one minute taken over the course of nearly four years. All of the features are sampled every minute, except for approximately $1.25\%$ of the records which have no measurements. We simply fill in these missing values by repeating the record that immediately precedes it. Since the data set does not have natural sparse and decay features, we have to artificially create them, which is described in the Appendix.

For the experiments described in this section, we use sequences covering two hours worth of power data. The recurrent part of our network is composed of two stacked RNN cells. We compare using two architectures: one has STLSTM at the bottom layer and LSTM at the second layer, and the other has TLSTM at the bottom layer and LSTM at the second layer. Experiments have shown that using two layers is optimal. We use a standard LSTM cell as the second layer instead of STLTSM or TLSTM cells because we assume the time and sparsity information is encoded in the output of the base layer that is fed into the second layer. These all have a hidden dimension of 64, which is also the size of the single embedding layer for the static standard features. Using a same hidden layer size means that STLSTM cells have more parameters than the TLSTM cells, but preliminary experiments on TLSTM cells showed this is the optimal architecture for performance during inference. For all results the static features are present, unless it is stated otherwise.

The first metric we investigate is the relative performance of the STLSTM cell versus the standard TLSTM cell versus sparsity of individual sparse features. For this we use a dense layer as the aggregation function in the STLSTM cell. For each of the seven features, we set that feature to be sparse with a sparse ratio ranging from $0.01$ to $0.15$ and the remaining six to be dense. For the TLSTM model, the sparse features are input at every time step as dense features. Figure 5 shows the relative performance on F1 scores between the TLSTM and STLSTM models for both types of record subsampling, with percentages greater than $0$ meaning the STLSTM performs better. The F1 score of STLSTM, which corresponds to the denominator, is approximately 0.7 but depends on which feature is made to be sparse.

A common characteristic for all series is that the relative performance of the STLSTM cell increases as the sparsity increases, but only to a certain point where it then maintains or slightly drops. Further, STLSTM offers a larger improvement in the case of the group sampling in most cases. As can be seen in Figure 5, the relative performance of each model depends on the feature we are treating as sparse. It is not surprising that the feature with the largest dependence on sparsity is the voltage

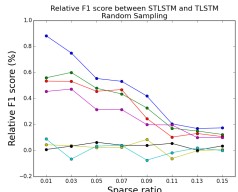 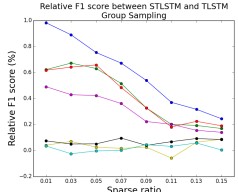 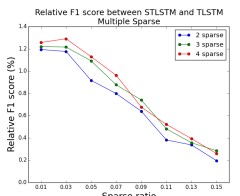 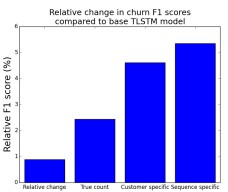

Figure 5: Relative change in F1 score between STLSTM and TLSTM cell. For single sparse features, random sampling is first and group sampling second. Third is 2, 3, and 4 sparse features using group sampling. Fourth is relative performance on different sparse subsets of features for the churn data set.

variable itself (represented by blue), since this is the feature we are making predictions on. The features with the smallest relative performance change are found to have the smallest response to sparsity as well, which may indicate that they have less predictive power and thus it is not surprising that the difference between the TLSTM and STLSTM is also small.

Further than looking at individual sparse features, we also investigate groups of sparse features with different levels of sparsity. We report the results on groups of sparse features with size 2, 3, and 4 since it was shown that 3 of the features are not affected by sparsity. Figure 5 also shows the relative performance for STLSTM and TLSTM models for groups of sparse features that were found to have the largest effect at a given group size and sparsity. Unsurprisingly, the characteristics of the series are the same as in the analysis of single features. Using more sparse features does increase the relative performance of the STLSTM cell to a point, but for this dataset where the features are closely related to one another the effectiveness is limited.

In addition to focusing on the relative performance between the TLSTM and STLSTM cells, we also consider the performance of different aggregation methods $\mathcal{L}$ within the STLSTM cell. In particular, we compare using a dense layer, an averaging aggregation, and a maximum aggregation at four different levels of sparsity. The aggregation method clearly does not matter if there is only one sparse feature, and so for this comparison we must use sets of sparse features. In particular, we report results using the set of 3 sparse features that was found to give the largest relative improvement for STLSTM. The results are summarized in Table 1a based on group sampling datasets.

As can be seen in Table 1a, the dense layer aggregation performs as well or better than the other two methods at every level of sparsity. The average and maximum aggregation have similar performance, perhaps slightly leaning toward the average method at low sparsity and the maximum method for higher sparsity.

Table 1: Aggregation and static feature analysis.

(a) F1 scores for STLSTM model with different aggregation methods for different levels of sparsity.

| Sparse ratio | Dense layer | Average | Max |
|---|---|---|---|
| 0.03 | **0.668** | 0.658 | 0.659 |
| 0.07 | **0.675** | 0.664 | 0.666 |
| 0.11 | **0.683** | 0.674 | 0.672 |
| 0.15 | **0.693** | 0.685 | 0.681 |

(b) Averaged relative F1 score change after incorporating one or both types of static features on group sampling datasets.

| Sparse ratio | Static standard | Static decay | Both |
|---|---|---|---|
| 0.03 | 1.56 | 0.67 | **2.01** |
| 0.07 | 1.48 | 0.58 | **1.92** |
| 0.11 | 1.45 | 0.51 | **1.76** |
| 0.15 | 1.44 | 0.48 | **1.69** |

Finally, we study the effect of adding both types of static features to the model. Table 1b shows the average relative improvements in F1 score for adding in one or both types of static features for different subsets of sparse features with varying levels of sparsity. It is clear from Table 1b that both types of features improve the performance of the model, with the static standard features having a larger effect. This larger effect may be specific to the features of the power consumption data, but generally both types of features improve performance when properly integrated into the model. With respect to sparsity, both types of static features improve the model more as the features become more sparse, with the static decay features being more dependent on sparsity.

It is worth discussing the impact of the added complexity of the STLSTM cell on training time. With STLSTM, as the number of sparse features grows, so too does the number of parameters in the model. This makes training slower per epoch, but it is mitigated in wall-clock training time by the fact that it takes fewer epochs to reach convergence. Using standard TLSTM, an epoch of training time takes about 10 seconds with the given sequence parameters for the datasets. When using four sparse features, this increases to about 18 seconds per epoch. However, the STLSTM cell generally reaches convergence in 40-50 epochs while the TLSTM cell requires 70-80 epochs.

## 4.2 REAL WORLD CHURN DATA SET

In addition to this constructed dataset, we also use a real world proprietary dataset with naturally sparse features for churn prediction within a future time span. At each time step, there is an event embedding that is treated as a dense feature. In addition to this, there are approximately 60 additional sequence state features. These state features have varying levels of sparse ratios, ranging from 0.03 to 0.6. We experiment with different subsets of dense and sparse features using the sequence state features. Additionally, we use two decay features, one static standard feature, and one static decay feature. For this dataset we use a training set with 700,000 samples and validation/test sets with 200,000 samples each. Sequence length ranges from 1 to 150, but the samples are not evenly distributed by sequence length. The distribution is exponential, with a large portion of sequences on the lowest end of that range and the mean is approximately 40.

For this data set, we compare two stacked architectures composed of three stacked RNN cells. As before, the bottom layer uses either STLSTM or TLSTM, and the higher levels use standard LSTM cells. When multiple sparse features are present, we use a dense layer as the aggregation method since it performs best. All results are given relative to the standard TLSTM model with all 60 additional sequence state features treated as dense features, as this was found to outperform an all LSTM model by approximately 2% under the all dense setting.

Since this data set is naturally sparse, we do not tune the sparsity, but instead investigate the relative performance for different subsets of sparse and dense features. There are four qualitative feature subsets we primarily study. The first sparse subset contains features that relate to relative changes, e.g. a customer can upgrade/downgrade his service to a different tier. The second sparse subset includes true counts for feature values, e.g. the total number of people associated with an account. These two subsets are the most sparse and have average sparsity of 0.065. The other two feature subsets contain features relating customer specific (e.g. a customer relocates) and sequence specific information (e.g. the number of emails to customer service). The relative performance for these sparse feature subsets is found in Figure 5. We observe that using only relative change features gives the smallest performance gain. This is not surprising because in this case a repeated value of 0 has meaning and the features are specifically designed to capture changes, somewhat mitigating the intended effect of the STLSTM cell. Using the raw counts does give a larger performance increase.

Both customer and sequence specific subsets show the largest improvement, indicating that these sub groups are responsive to STLSTM architecture. It is interesting that these two subgroups do not necessarily contain only the most sparse features, but a mix between sparsity ratios roughly from 0.03 to 0.3. This suggests that in real applications with naturally sparse features there is a large range of sparsity that can occur in the sparse sub group of features.

## 4.3 FUTURE WORK

For future work, it would be interesting to incorporate even more feature types than the five covered in this work. One in particular is a feature type that gives time information looking forward in the sequence. All features in this work use time information related to past events, but there are cases that can benefit from the utility of incorporating future knowledge when available. One example of this is the time to the prediction from the current time step so the network can have direct knowledge of its absolute time location in the sequence.

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

## 5 APPENDIX

The power consumption dataset is one long sequence, and so we break this down into many sequences. We create the sequences based on three parameters: sample time $T_s$, shift window, and sequence length $T$. To create the first sequence we first set the start index to the first record. A full sequence is created by selecting $T_s$ records beginning at the start index. We further subsample from the $T_s$ records in order to obtain final sequences with decay features in two ways. In the first way we randomly sample individual records. The first record in the full sequence is always kept, but after the first record we randomly drop records based on Bernoulli until there are $T$ remaining. In the second way, we create groups of 5 records. We start with the first 5 records from the start index. Next we focus on records from start index + 8 to start index + $T_s$. We select a random record and the surrounding 4 records. We then repeat the procedure until we have $T$ selected records. We finish with groups of 5 consecutive records that do not overlap. In either way we sample, this means the sampling rate is no longer a uniform one minute, and we use the time between two adjacent events to be the decay feature. The seven electrical features are dense features present at every time step in the sequence. The start index is then increased by the shift window and the process continues until we run through the entire dataset. If the shift window is less than the sample time, then there can be some overlap in the sequences. We allow these overlaps in the training set, but remove any sequences in the validation and test sets with subsequences duplicated in the training set.

Once the sequences are all created, we can create sparse features on any subset of the seven dense features. This is done by choosing a sparse ratio for each feature in the sparse subset, where the sparse ratio is the ratio of feature values that are kept in each sequence. We keep feature values present with uniform probability equal to the sparse ratio for that feature. If a feature value is not kept at a time step, its mask value is set to 0 at the time step. This allows us to easily experiment with different numbers of sparse features and the extent of the sparsity.

Both types of static features are also used for this dataset. There is one static decay feature, and it is the time elapsed between the last event of the sequence and the prediction time for that sequence, which is $T_s$ events from the time of the first event. There are three categorical static standard features, and these are the day of the week, day of the month, and time of day (morning, afternoon, evening, night). Each of the sequences we use spans two hours or less, and we assign the static standard feature values based on the first event in the sequence. Figure 6 depicts these concepts.

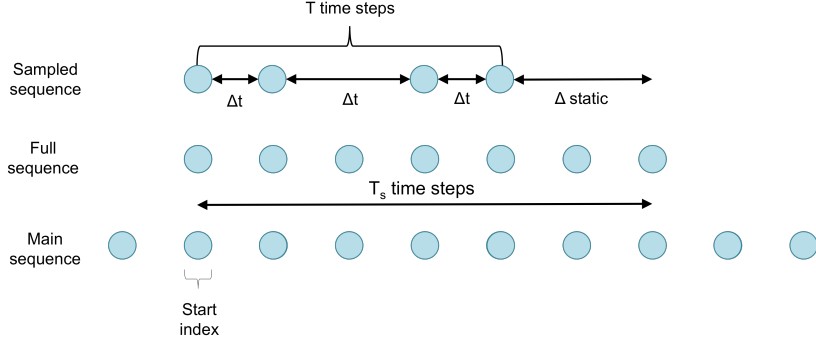

Figure 6: The creation of sequences from the electric power consumption dataset.

The training target for the power consumption dataset is a multi-class classification target. We predict whether the average voltage over some time interval after the prediction time of a sequence stays within half a standard deviation, increases by more than half a standard deviation, or decreases by more than half a standard deviation when compared to the value averaged over the time span of the sequence measured from the first event to $T_s$ time steps after the first event. The standard deviation value here is taken over the entire training set. This is an unbalanced classification task. While the exact majority class percentage varies depending on the way the sequences are created, it falls between $65\%$ and $80\%$ for our experiments. For the experiments described in Section 4.1 we use two hours of data corresponding to $T_s = 120$. We also use a shift window of 30 and set $T = 50$.

