# OpenReview forum: "Unified recurrent network for many feature types"
_ICLR.cc/2020/Conference — Reject_

### Official Review · AnonReviewer2 · 2019-10-29
**Official Blind Review #2**

**Rating:** 1

**Review:**

This submitted manuscript is exactly the paper (bearing no difference) that was submitted to ICLR 2019 and also rejected.
This submitted manuscript is exactly the paper (bearing no difference) that was submitted to ICLR 2019 and also rejected.
This submitted manuscript is exactly the paper (bearing no difference) that was submitted to ICLR 2019 and also rejected.
This submitted manuscript is exactly the paper (bearing no difference) that was submitted to ICLR 2019 and also rejected.
This submitted manuscript is exactly the paper (bearing no difference) that was submitted to ICLR 2019 and also rejected.
This submitted manuscript is exactly the paper (bearing no difference) that was submitted to ICLR 2019 and also rejected.

**Experience Assessment:**

I have read many papers in this area.

**Review Assessment: Checking Correctness Of Derivations And Theory:**

I assessed the sensibility of the derivations and theory.

**Review Assessment: Checking Correctness Of Experiments:**

I assessed the sensibility of the experiments.

**Review Assessment: Thoroughness In Paper Reading:**

I read the paper at least twice and used my best judgement in assessing the paper.

---

### Official Review · AnonReviewer3 · 2019-10-30
**Official Blind Review #3**

**Rating:** 1

**Review:**

I have checked the paper that AnonReviewer1 pointed out and verified that the two papers are identical. It would be better to reject the paper.

I would also appreciate authors to consider the comments from the previous ICLR-19 reviews and improve the paper.

I also think that the desk rejection would be an appropriate decision for this paper.

At least it would be good to add the following papers as pointed by one of review last year.
- Pham, T., Tran, T., Phung, D., & Venkatesh, S. (2016, April). DeepCare: A deep dynamic memory model for predictive medicine. In Pacific-Asia Conference on Knowledge Discovery and Data Mining (pp. 30-41). Springer, Cham.
- Koutnik, J., Greff, K., Gomez, F., & Schmidhuber, J. (2014). A clockwork RNN. arXiv preprint arXiv:1402.3511.
- Chen, C., Kim, S., Bui, H., Rossi, R., Koh, E., Kveton, B., & Bunescu, R. (2018, October). Predictive Analysis by Leveraging Temporal User Behavior and User Embeddings. In Proceedings of the 27th ACM International Conference on Information and Knowledge Management (pp. 2175-2182). ACM.

**Experience Assessment:**

I have published in this field for several years.

**Review Assessment: Checking Correctness Of Derivations And Theory:**

I did not assess the derivations or theory.

**Review Assessment: Checking Correctness Of Experiments:**

I did not assess the experiments.

**Review Assessment: Thoroughness In Paper Reading:**

I made a quick assessment of this paper.

---

### Official Review · AnonReviewer4 · 2019-11-02
**Official Blind Review #4**

**Rating:** 1

**Review:**

This paper proposes Sparse Time LSTM, an extension of TLSTM to handle (the most importantly) the sparse features which is updated infrequently, and also the static features which does not change across the time. For sparse features, authors proposed an lazy update mechanism where the memory state is updated only at the time of a sparse feature update (i.e. when $m_{tk} = 1$), and then the hidden state / output gate are updated by aggregating over the hidden state proposals for all sparse features (using either average-/max-pooling or a MLP). For static features, the authors proposed inputing the feature embeddings only at the final time steps. The proposed method can also handle decay features by using the decay mechanism that already exists in TLSTM (although the authors proposed an extension to generalize the mechanism to vector input). The authors tested the proposed method on a synthetic dataset based on UCI electronic power consumption data, and a proprietary dataset for churn prediction.

I vote to reject the paper in its current form. I believe the paper is tackling an important task (extending neural network methods for sparse and missing time-series data), and I find the proposed methods intuitive and the experiment evaluations interesting. However, I find the technical description of the proposed method insufficient, in the sense that (1) there does not seem to be explanation about how the proposed mechanism is better than the previous method (e.g. intuition or theoretical justification of why handling sparse update using the TLSTM approach is inferior), or (2) no justification about the particular choice of the model (e.g. in Section 3.2, why choose this particular form of g)? Also, the experiment section does not seem to contain sufficient detail to be reproducible (e.g., how many simulation runs were conducted?), and the reported results do not contain standard error information which makes it difficult to decide the significance of the improvement. The organization of the paper can also be improved (see Major comments). In summary, the paper in its current form does not yet reach the ICLR standard due to the lack of clarity in its technical description, and lack of sufficient detail in experiment results to be convincing. This paper can benefit greatly with sufficient justification of its choice of model mechanisms, a more careful explanation of the advantage of its approach to sparse feature relative to previous approaches, and also a self-contained description of the experiment procedures and standard error information for the experiments.

Although the proposed method is of limited novelty (a straightforward extension to TLSTM) and the proposed methods for static/decay features feels somehow trivial, I feel this work is still interesting since it discusses an interesting topic. For future improvements, in addition to incorporating reviewers' comments from past submissions,  authors can consider shaping it into an in-depth study of how LSTM approaches should best handle sparse features.



Major Comments:

The organization of the method section can be improved. Earlier descriptions of the TLSTM should become a separate section called Related Work or Background, or at least being clearly marked as previous work. The proposed STLSTM method, which is the major innovation of this paper, should be introduced in the beginning of the Method section (rather than being delayed to Section 3.3).


A more detailed explanation of experiment protocol is needed. What is group sampling, and how many times were the experiments repeated? On a related note, in Table 1a-1b, please also report the standard error of F1 scores across multiple runs. This is needed to access whether the improvements brought by dense layer / additional features are significant.

Minor Comments:

Please add equation numbers to all equations.

Equation on page 5, please explain explicitly what is being aggregated over by the aggregation function L. Is the hidden states for all sparse features?

The titles and labels for the experiment result figure (Figure 5) are too small to be readable. Also there's no legend for the first two figures. What does each colored line mean? My guess is they are corresponding to the seven features, however author should explain this clearly in the figure caption or in the text description.

**Experience Assessment:**

I have published one or two papers in this area.

**Review Assessment: Checking Correctness Of Derivations And Theory:**

N/A

**Review Assessment: Checking Correctness Of Experiments:**

I carefully checked the experiments.

**Review Assessment: Thoroughness In Paper Reading:**

I read the paper thoroughly.

---

### Decision · Program_Chairs · 2019-12-19

**Decision:**

Reject

**Comment:**

main summary: sparse time LSTM

discussions;
reviewer 4: technical description of the proposed method insufficient,
reviewer 2, 3: same paper sent to ICLR 2019 and rejected
recommendation: rejected, based on all reviewers comments